

PeerJ Hubs
Published on behalf of

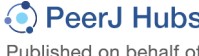
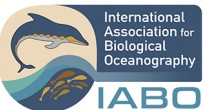

# Deep-ocean macrofaunal assemblages on ferromanganese and phosphorite-rich substrates in the Southern California Borderland

Michelle Guraieb[1], Guillermo Mendoza[1], Kira Mizell[2], Greg Rouse[1], Ryan A. McCarthy[3], Olívia S. Pereira[1] and Lisa A. Levin[1]

[1] Integrative Oceanography Division and Center for Marine Biodiversity and Conservation, Scripps Institution of Oceanography, University of California, San Diego, La Jolla, California, United States
[2] USGS Pacific Coastal and Marine Science Center Santa Cruz, Santa Cruz, CA, USA, Santa Cruz, California, United States
[3] Marine Physical Laboratory, Scripps Institution of Oceanography, University of California San Diego, La Jolla, California, United States

Corresponding author
Michelle Guraieb,
mguraieb@ucsd.edu

## ABSTRACT

Mineral-rich hardgrounds, such as ferromanganese (FeMn) crusts and phosphorites, occur on seamounts and continental margins, gaining attention for their resource potential due to their enrichment in valuable metals in some regions. This study focuses on the Southern California Borderland (SCB), an area characterized by uneven and heterogeneous topography featuring FeMn crusts, phosphorites, basalt, and sedimentary rocks that occur at varying depths and are exposed to a range of oxygen concentrations. Due to its heterogeneity, this region serves as an optimal setting for investigating the relationship between mineral-rich hardgrounds and benthic fauna. This study characterizes the density, diversity, and community composition of macrofauna (>300 µm) on hardgrounds as a function of substrate type and environment (depth and oxygen ranges). Rocks and their macrofauna were sampled quantitatively using remotely operated vehicles (ROVs) during expeditions in 2020 and 2021 at depths above, within, and below the oxygen minimum zone (OMZ). A total of 3,555 macrofauna individuals were counted and 416 different morphospecies (excluding encrusting bryozoans and hydrozoans) were identified from 82 rocks at depths between 231 and 2,688 m. Average density for SCB macrofauna was $11.08 \pm 0.87$ ind. 200 cm$^{-2}$ and mean Shannon-Wiener diversity per rock (H$'_{[\text{loge}]}$) was $2.22 \pm 0.07$. A relationship was found between substrate type and macrofaunal communities. Phosphorite rocks had the highest H$'$ of the four substrates compared on a per-rock basis. However, when samples were pooled by substrate, FeMn crusts had the highest H$'$ and rarefaction diversity. Of all the environmental variables examined, water depth explained the largest variance in macrofaunal community composition. Macrofaunal density and diversity values were similar at sites within and outside the OMZ. This study is the first to analyze the macrofaunal communities of mineral-rich hardgrounds in the SCB, which support deep-ocean biodiversity by acting as specialized substrates for macrofaunal communities. Understanding the intricate relationships between macrofaunal assemblages and mineral-rich substrates may inform effects from environmental
disruptions associated with deep-seabed mining or climate change. The findings contribute baseline information useful for effective conservation and management of the SCB and will support scientists in monitoring changes in these communities due to environmental disturbance or human impact in the future.

# INTRODUCTION[1]

[1] Portions of this text were previously published as part of a thesis (https://escholarship.org/content/qt4tz9r4fh/qt4tz9r4fh.pdf?t=sbf1ne).

The deep ocean (>200 m deep) is the largest habitable space on Earth and it remains the least explored and understood area of the ocean. With an average depth of 3,800 m, the ocean consists mostly of deep water, which represents over 95% of the volume on Earth that is available for living organisms to thrive (*Danovaro et al., 2020*). The deep ocean plays a vital role in regulating our climate and providing essential services and resources to humanity (*Thurber et al., 2014*; *Baker, Ramirez-Llodra & Tyler, 2020*). However, deep-ocean ecosystem services provided to humankind are under pressure from human activities that increasingly impact the natural functions that occur in the ocean (*Baker, Ramirez-Llodra & Tyler, 2020*). Advancing our knowledge of the deep ocean through baseline studies supports the development of conservation initiatives and effective marine ecosystem management strategies. This is especially urgent as cumulative impacts from climate change to deep-seabed mining pose challenges for the proper management of the deep ocean (*Levin et al., 2016*; *Baker, Ramirez-Llodra & Tyler, 2020*).

Two deep-ocean mineral types being considered for their resource potential are ferromanganese (FeMn) crusts and phosphorite rocks (*Hein et al., 2013*, *2016*). FeMn crusts, which were first considered as a potential resource for cobalt in the early 1980s, are also enriched with metals, such as copper, nickel, and manganese, which are used in electric car batteries and other technologies (*Hein et al., 2013*). FeMn crusts precipitate from seawater and are typically found in open ocean areas with low organic carbon content and low sedimentation rates (*Hein et al., 2013*; *Usui et al., 2017*). These crusts are found across a broad range of depths (400–7,000 m) on seamounts, ridges, and plateaus and form within a variety of seawater oxygen concentrations (*Hein et al., 2013*; *Mizell et al., 2020*).

Marine phosphorites occur in the Pacific and Atlantic Oceans along the western continental margins at depths shallower than 2,500 m in upwelling areas, on seamounts, and in lagoon deposits (*Hein et al., 2016*). Some shallower, nearshore occurrences of these phosphorous-rich rocks are of interest to the mining industry as a source of macronutrients for fertilizers used in agriculture (*Pacific Coastal and Marine Science Center (PCMSC), 2022*); and rare earth elements, which have been studied as a potential secondary ore (*Hein et al., 2016*).

Extraction of these mineral-rich geological features will affect deep-ocean biodiversity, and research regarding potential impacts from deep-seabed mining to the health of the global ocean is needed (*Levin et al., 2016*). FeMn crusts and phosphorite rocks are

inherently interwoven with the life of deep-ocean fauna as they cover miles of the seafloor where animals live and biogeochemical processes fundamental to the overall balance of ocean ecosystems occur (*Jones, Amon & Chapman, 2018*). Currently, the relationship between macrofaunal communities and mineral-rich hard substrates is not well understood beyond provision of attachment sites or physical habitat (*Schlacher et al., 2014*).

Biodiversity and species abundance in the deep ocean are responsible for key ecological functions, such as nutrient cycling, bioturbation, connectivity, primary and secondary production, respiration, habitat, and food supply (*Levin et al., 1997*; *Miller et al., 2012*; *Thurber et al., 2014*; *Le, Levin & Carson, 2017*). These ecological functions translate into provisioning, regulating, and cultural services (*Le, Levin & Carson, 2017*). Provisioning services include fisheries (*Clark et al., 2010*), industrial agents (*Mahon et al., 2015*), and pharmaceuticals and biomaterials (*Liu et al., 2013*). Regulating services involve climate regulation (*Jiao et al., 2010*), biological control, and waste absorption (*Pham et al., 2014*). Cultural services encompass educational, aesthetic, existence, and stewardship benefits (*Thurber et al., 2014*). The loss of biodiversity leads to a decline in these important functions and services on which we rely (*Danovaro et al., 2008*). Furthermore, the biodiversity in the deep ocean is a crucial component of the resilience of these ecosystems, contributing to their ability to withstand the effects of anthropogenic disturbance (*Thurber et al., 2014*). For example, an ecosystem with higher genetic variability will exhibit a range of tolerance levels toward disturbance, making it more adaptable and resilient (*Oliver et al., 2015*).

Deep-water fishing, climate change, oil and gas extraction, pollution, and deep-seabed mining are some of the main potential stressors to deep-ocean ecosystems (*Ramirez-Llodra et al., 2011*). According to *Levin et al. (2016)*, mining of FeMn crusts and phosphorite rocks would remove currently living structure-forming organisms that provide habitat and food for other smaller fauna (*Buhl-Mortensen et al., 2010*), resulting in the loss of heterogeneity and therefore driving a decline in biodiversity. Moreover, the consequences of climate change, including oxygen loss, have the potential to amplify the adverse impacts stemming from mining operations (*Le, Levin & Carson, 2017*). The emerging consensus is that more scientific information is needed to inform regulations and decision-making regarding these deep-sea ecosystems and their disturbance (*Levin et al., 2016*; *Jones, Amon & Chapman, 2018*; *Montserrat et al., 2019*).

## Oxygen minimum zone of the Southern California Borderland

The Southern California Borderland (SCB) offers a unique environment to study the relationship between mineral-rich hardgrounds and the benthic fauna that live on them. A variety of geological features (*e.g.*, banks, ridges, knolls, escarpments and seamounts) and environmental conditions (*e.g.*, low oxygen, various depths, varying food supply, and temperature ranges) add to the heterogeneity of the region and make it a suitable habitat for many marine species that inhabit hard substrates. The SCB exhibits characteristics that allow for a well-formed oxygen minimum zone (OMZ) at bathyal depths (400–1,100 m in the case of this study) because it is located on the eastern boundary of the Pacific Ocean

Basin, where upwelling acts as one of the drivers of oxygen depletion (*Gooday et al., 2010*). Equatorward winds blowing along the coast in the SCB underpin the upwelling of nutrients from depths of 200 m, supporting high primary productivity (*Checkley & Barth, 2009*). High productivity in shallow waters leads to large amounts of organic matter sinking to deeper depths where bacteria use oxygen to decompose organic particles, further driving oxygen depletion (*Levin, 2003*).

Ocean deoxygenation is a phenomenon characterized by the reduction of dissolved oxygen content in the ocean due to human activities, primarily the addition of nutrients and global warming (*Breitburg et al., 2018*; *Oschlies et al., 2018*). Warming has contributed to the expansion of OMZs, areas where oxygen concentrations are $<0.5$ ml $L^{-1}$ or $<22$ μM kg$^{-1}$ (*Levin, 2003*; *Stramma et al., 2010*). Significant loss of oxygen off Southern California is also attributed to the strengthening of the California Undercurrent, which brings warm, salty, low oxygen water up from the equator (*Bograd et al., 2015*). The expansion of OMZs and oxygen reduction can compress the habitat of marine species, which may trigger a variety of biological responses (*Stramma et al., 2010*). In sediments, macrofaunal densities are lowest in the core of the OMZ (*Levin, 2003*), and diversity decreases with declining oxygen concentrations (*Levin, Huggett & Wishner, 1991*; *Levin et al., 2002*; *Gooday et al., 2010*), which we expected to see for hard-ground communities in the OMZ of the SCB.

## Deep-ocean faunal studies and their relevance

During the past three decades, technological advancements have allowed scientists to study the diversity, ecology, and surrounding environment of deep-ocean macrofauna using multicores (*De Smet et al., 2017*), submersibles (*Li, 2017*; *Dong et al., 2021*), and ROVs (*Schlacher et al., 2014*). Various studies have explored the relationship between benthic faunal communities and the substrate on which they live (*Gage & Tyler, 1991*; *Gooday et al., 2010*; *Vanreusel et al., 2010*; *Schlacher et al., 2014*; *Simon-Lledó et al., 2019*; *Pereira et al., 2022*); however, most studies that examine this relationship have focused on chemosynthetic ecosystems (*Baco & Smith, 2003*; *Levin et al., 2015*; *Levin, Mendoza & Grupe, 2017*; *Bourque et al., 2017*; *Pereira et al., 2021*, *2022*) and deep-ocean sediments (*Wei et al., 2012*; *Baldrighi et al., 2014*; *Leduc et al., 2015*; *De Smet et al., 2017*; *Dong et al., 2021*). Those studies undertaken on non-reducing, hard substrates have mainly examined the characteristics of the megafauna community (*Clark, 2011*; *Grigg et al., 1987*; *Amon et al., 2016*; *De Smet et al., 2021*; *Vlach, 2022*; *Yan et al., 2024*). Studies of macrofaunal assemblages on mineral-rich substrates are limited and often examine the fauna of associated sediments (*e.g.*, *Leduc et al., 2015*; *Chuar et al., 2020*). Research on abyssal plains with polymetallic nodules has focused largely on foraminifera (*Mullineaux, 1987*, *1989*; *Veillette et al., 2007*), meiofauna (*Pape et al., 2021*), megafauna (*Tilot, 2006a*, *2006b*; *Kersken, Janussen & Martínez Arbizu, 2018*; *De Smet et al., 2021*), or macrofauna in the sediments surrounding the nodules (*De Smet et al., 2017*; *Chuar et al., 2020*; *Lins et al., 2021*). *Grischenko, Gordon & Melnik (2018)* addressed macrofauna on nodules in the Clarion Clipperton Zone, focusing exclusively on Bryozoa only, and the one study of macrofauna on FeMn crusts was only qualitative (*Toscano & Raspini, 2005*). Our

analysis is the first to examine the macrofaunal relationship with mineral-rich hard substrates in a quantitative analysis of density, diversity, and community composition in the SCB region.

The objective of this study is to understand the relationship of macrofaunal (>300 μm) assemblages to mineral-rich substrates in the SCB off the Pacific coast of the United States, and to other environmental factors. For substrates collected during two oceanographic expeditions (NA124 cruise in 2020 and FK210726 cruise in 2021), we characterize the macrofaunal (>300 μm) density, diversity, and community composition (hereafter community structure) of the mineral-rich and other hard substrates of the SCB. Specifically, we examine faunal association with a) various substrate types (FeMn crust, and phosphorite, basalt, and sedimentary rocks); and b) different environmental variables (oxygen and water depth). We hypothesize that (1) benthic community structure varies across different types of mineral-rich hard substrates, (2) macrofaunal communities in deeper waters are significantly less dense but more diverse than at shallower depths, and (3) macrofaunal assemblages within the OMZ exhibit lower densities and diversity compared to those in more oxygenated areas (above and below the OMZ).

These data provide baseline information that can inform decision-making processes and support management strategies for biodiversity in Southern California and in the deep ocean. Studying healthy ecosystems such as the SCB offers opportunities for comparative studies of areas currently under consideration for deep-seabed mining activities. Regions such as the West-Pacific seamounts hosting FeMn crust, as well as the continental margins of Mexico, Namibia, South Africa, and New Zealand with phosphorites are currently or have recently been considered for their economic potential, causing a need to understand the biodiversity of these mineral systems (*Levin et al., 2016*). Safeguarding and studying analogous ecosystems in regions not currently targeted by mining enterprises, such as the SCB, can contribute to the overarching objective of maintaining the integrity of ecosystems, their services, and functions. These data will inform ocean stakeholders, including the people of California, who are spiritually, culturally, and economically connected to the deep ocean (*California State Legislature, 2022*).

## MATERIALS AND METHODS[1]

### Study area and data collection

Rocks of different mineral types (Table 1) along with their biological community were collected in the SCB during two research expeditions aboard the E/V *Nautilus* (NA124-October 28 to November 6 of 2020) and R/V *Falkor* (FK210726-July 26 to August 6 of 2021) at the locations shown in Fig. 1. Rocks, with a surface area ranging from 273.43 to 1,963.87 cm$^2$, were collected using the remotely operated vehicles (ROVs), *Hercules* (onboard E/V *Nautilus*) and *SuBastian* (onboard R/V *Falkor*). *Hercules* was equipped with two manipulator arms, one high-definition video camera, LED lights, and a CTD (Conductivity-Temperature-Depth) sensor. *SuBastian* was equipped with two manipulators, two high-resolution video cameras, LED lights, and a CTD sensor.

**Table 1 Sites visited aboard E/V _Nautilus_ (NA124) and R/V _Falkor_ (FK210726) with the dive number, date sampled, site name, depth range of rocks collected and physical coordinates at the start of the dive.** Substrate type and number of rocks collected are shown for each dive.

| Cruise number | Site/Location (coordinates) | Depth range of rocks (m) | Average temperature (°C) | Oxygen range (µmol $L^{-1}$) | Basalt rocks | Phos-phorite rocks | FeMn crusts | Sedi-mentary rocks |
|---|---|---|---|---|---|---|---|---|
| NA124 | Patton Escarpment Central (33.0625 N, −120.1229 W) | 587–820 | 5.34 | 2.61–5.14 | 1 | 1 | | 3 |
| | San Juan Seamount upper flank (33.0390 N, −121.0050 W) | 691–1,129 | 4.39 | 2.79–14.04 | | | 5 | |
| | Northeast Bank (32.3168 N, −119.5981 W) | 553–1,132 | 5.15 | 2.68–15.95 | 2 | | 3 | |
| | Cortes Bank (32.4154 N, −119.2989 W) | 437–529 | 6.41 | 4.14–8.31 | | 5 | | |
| | Patton Ridge South (32.7384 N, −120.0125 W) | 562–726 | 5.40 | 2.55–3.96 | | 5 | | |
| | 40-Mile Bank (32.6003 N, −118.0283 W) | 658–1036 | 4.76 | 2.11–14.6 | | | 2 | 3 |
| | San Clemente Escarpment (32.6745 N, −118.1312 W) | 1,189–1,718 | 3.03 | 15.27–37.46 | 1 | | 4 | |
| | Osborn Bank Mesophotic Zone (33.3420 N, −119.0457 W) | 231–396 | 7.83 | 19.71–54.79 | 4 | | | |
| FK210726 | Hancock Bank (32.5450 N, −119.6761 W) | 319–594 | 6.54 | 9–36.48 | 6 | 1 | 1 | |
| | San Juan Seamount North (33.0333 N, −120.9999 W) | 1,138–1,442 | 3.29 | 26.7–36.48 | | | 8 | |
| | Patton Escarpment (32.4028 N, −120.1468 W) | 1,453–1,797 | 2.60 | 46.89–70.9 | | | 6 | |
| | Little Joe Seamount (31.8962 N, −120.0303 W) | 2,366–2,688 | 1.81 | 100.17–108.14 | | | 8 | |
| | Crespi Knoll (33.1014 N, −117.8853 W) | 443–525 | 7.08 | 12.99–33.11 | 5 | | | 1 |
| | Coronado Escarpment (32.6673 N, −117.4857 W) | 443–467 | 7.57 | 24.16–28.28 | | 7 | | |

## Sampling and at-sea processing

The ROVs collected rocks of four different types (Fig. 2) with the manipulator arms, maintaining each rock in its _in-situ_ orientation and jostling it as little as possible to preserve fauna settled on the substrate. Rocks were collected according to whether they fit into the biobox compartments (22 × 22 × 30 cm or 45 × 22 × 30 cm) and whether they could be lifted readily (not cemented to larger boulders) or could be broken off. Phosphorites and FeMn crusts were targeted specifically but other available substrate types (basalt, sedimentary rocks) were collected haphazardly. Each rock was placed into its own isolated biobox compartment on the ROV to avoid cross-contamination or loss of fauna during transport. At each rock collection location, the CTD attached to the ROV obtained measurements of temperature, salinity, pressure (depth), and oxygen concentrations. The rock substrates were processed quantitatively for their associated biological community. Every rock was photographed onboard ship on six sides with a scale and label. All the visible organisms were removed using forceps and kept in crystalizing dishes with cool seawater. The residual water contained in each biobox compartment holding individual

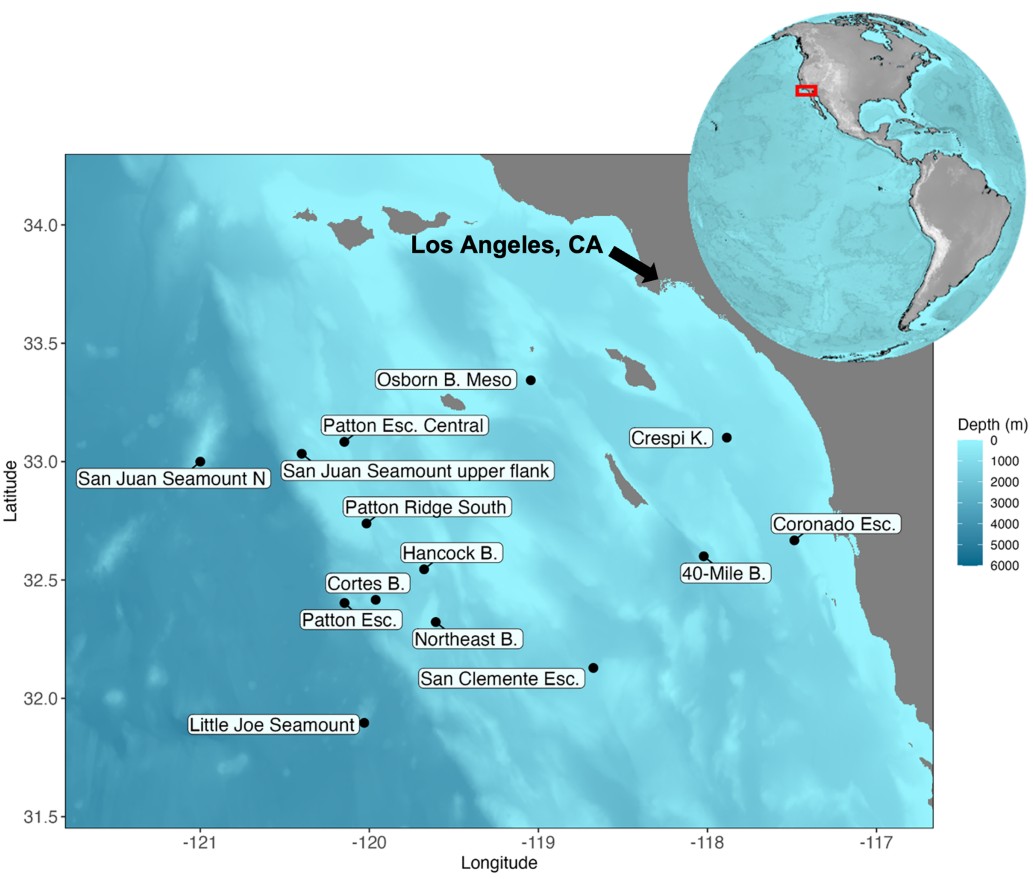

**Figure 1** **Map of the Southern California Borderland and the locations visited during the expedition aboard E/V *Nautilus* in 2020 (NA 124) and R/V *Falkor* in 2021 (FK210726).** Eight sites (Patton Esc. = Patton Escarpment, S.J. Seamount Upper Flank = San Juan Seamount Upper Flank, Northeast B. = Northeast Bank, Cortes B. = Cortes Bank, Patton Ridge South, 40-Mile B. = 40-Mile Bank, San Clemente Esc. = San Clemente Escarpment, Osborn B. Meso = Osborn Bank Mesophotic Zone) were visited on NA 124 and seven sites (Hancock B. = Hancock Bank, S.J. Seamount North = San Juan Seamount North, Patton Esc. = Patton Escarpment, L.J. Seamount = Little Joe Seamount, Crespi K. = Crespi Knoll, Coronado Esc. = Coronado Escarpment) were visited on FK210726. Map showing the study area with coordinates in degrees north latitude (°N) and west longitude (°W). The map was plotted in R software using ETOPO 2022 database hosted on the NOAA website in the public domain (*NOAA National Centers for Environmental Information, 2022*).

rocks was washed through a 0.3 mm mesh to collect the macrofauna, and the retained biota were preserved in ethanol. Then, the rocks were wrapped in a monolayer of aluminum foil, which was later weighed to obtain approximate surface area (as in *Levin et al., 2015*). Each rock was left overnight in a seawater bucket at room temperature to allow the remaining fauna to crawl away or fall out of the rock's crevices. Then, the water in each bucket was sieved again to recover the fauna, and these were combined with the previous day's collections from the same rock.

## Substrate identification

At the U.S. Geological Survey (USGS) laboratory, rock samples were cut along their longest axis using a diamond blade, and the cut face was described in detail regarding apparent

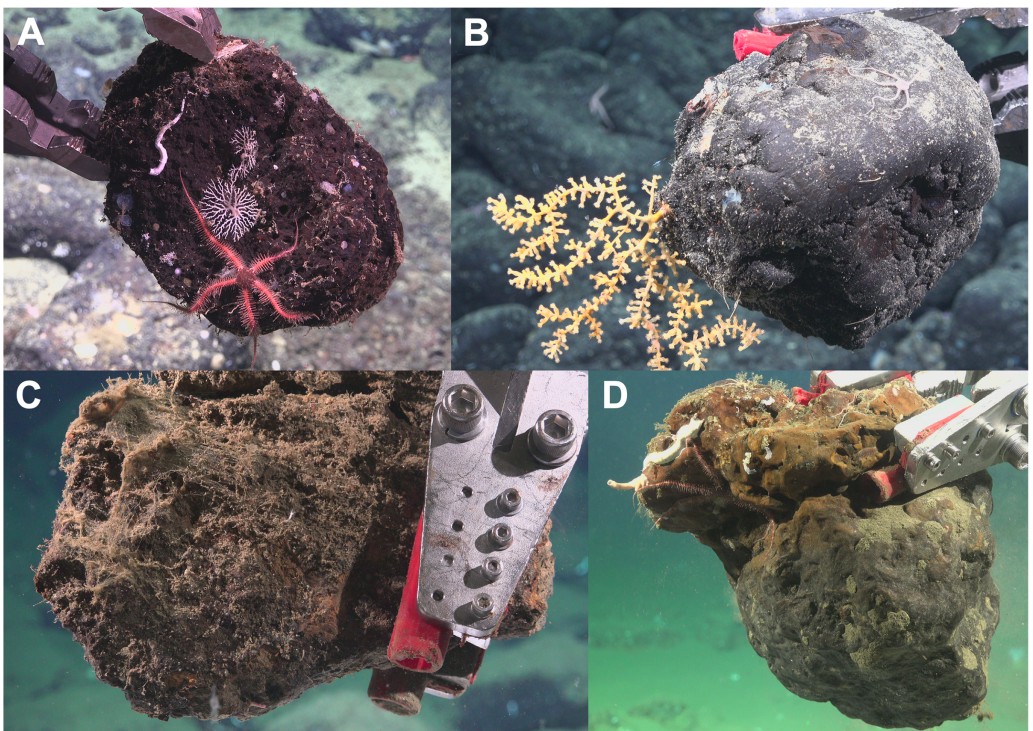

**Figure 2 Photos of each substrate type upon collection.** (A) Basalt (Hancock Bank–SCB-026; surface area = 978.12 cm$^2$); (B) Ferromanganese (FeMn) crust (San Juan Seamount North–SCB-077; surface area = 945.31 cm$^2$); (C) Sedimentary (Crespi Knoll–SCB-218; surface area = 610.93 cm$^2$); and (D) Phosphorite rock (Coronado Escarpment–SCB-329; surface area = 1,301.56 cm$^2$). Image credit: Schmidt Ocean Institute.

mineral type, stratigraphy, texture, and size. FeMn crusts are identifiable by color and morphology; they are black precipitates occurring on hard substrate of a different rock type (often basalt or other volcanic rock). Phosphorites are typically smooth, shiny, and dense; however, in some cases, it was difficult to tell if a sample was carbonate or phosphorite. To confirm mineral type when ambiguous, representative slabs were cut from each sample for crushing and powdering. The powdered sample was then analyzed by X-ray diffraction (XRD) to determine the presence of carbonate fluorapatite (phosphorite), calcite (carbonate or limestone), clay minerals (mudstone), or volcanic minerals (*e.g.*, feldspar). XRD data were produced by a Panalytical X'Pert3 X-ray diffractometer with CuKα radiation and graphite monochromator. The first and primary measurement for all samples was collected every 0.02 °2 theta between 4 ° and 70 °2 theta at 40 kV and 45 mA. Diffraction peaks from the digital scan data were identified using Phillips X'Pert High Score software, and mineral patterns were matched to patterns from the ICDD PDF4+ database.

## Laboratory processing and data synthesis

The surface area of collected rocks was obtained by weighing the aluminum foil wrap pieces using a top-loading balance. The foil weight was divided by the weight of 1 cm$^2$ of

foil to determine the rock surface area in $cm^2$. Densities below are expressed as the number of organisms per 200 $cm^2$.

The macrofauna preserved in ethanol at sea was re-sieved in the lab using a 0.3 mm mesh and sorted under a dissecting microscope at $12\times$ magnification. Taxonomic identification was done to the lowest taxonomic level of identification possible using morphological characteristics. Only 143 of the 418 taxa in this study were assigned a specific genus and, of those, 61 were given a species name; the remainder were identified to their lowest taxonomic level possible and designated as morphospecies. Encrusting Bryozoa were quantified by analyzing photographs capturing each profile of every rock at the time of collection aboard the ship. Voucher specimens of some macrofaunal morphospecies have been deposited to the Scripps Institution of Oceanography Benthic Invertebrate Collection (SIO-BIC).

## Univariate analysis

The dataset used to calculate densities included encrusting Bryozoa and Hydrozoa specimens. Total densities per sample were tested for normality using a Shapiro-Wilk test. A square-root transformation was applied to generate a normal distribution. Then, Bartlett's test was used to assess if the variances were homogeneous. A Kruskal-Wallis test followed by a Dunn's test using Benjamini-Hochberg adjustment (*Benjamini & Hochberg, 1995*) was performed when comparing densities across substrate types because the assumption of homogeneity was not met. We utilized regression analysis to explore how variations in oxygen concentration and depth correlate with changes in density, and to explore the correlation between surface area and macrofaunal density and diversity.

Data excluding encrusting Bryozoa and Hydrozoa (not identified to species level) were used to calculate diversity across various substrate types and environmental variables using Shannon-Weiner diversity index (H′), evenness (J′), rarefaction diversity $ES_{(20)}$ and species richness (S). H′ ($H'_{[log e]}, H'_{[log 10]}$), J′, and S were calculated using functions "diversity," "evenness," and "Estimate," respectively, from the *vegan* (*Oksanen et al., 2022*) package in R software (*R Core Team, 2022*). $ES_{(20)}$ rarefaction values and Shannon-Weiner diversity index (H′) were tested for normality using Shapiro-Wilk test and then a Cochran test was used to assess if the variances were homogeneous. To determine if there were any statistically significant differences within $ES_{(20)}$ values and Shannon-Weiner diversity index (H′) across substrate, a Kruskal-Wallis test followed by Dunn's tests using Benjamini-Hochberg adjustment was performed due to the variance of homoscedasticity. We utilized regression analysis to explore how variations in oxygen concentration and depth correlate with changes in diversity. All statistical analyses were performed in R software (*R Core Team, 2022*) using the packages *vegan* (*Oksanen et al., 2022*), *outliers* (*Komsta, 2022*) and *car* (*Fox & Weisberg, 2019*).

## Multivariate analysis

A multivariate analysis was performed to provide a measure of the dissimilarity of macrofaunal community composition between rock samples across different substrates. Total macrofaunal counts, excluding encrusting Bryozoa and Hydrozoa (not identified to

species level) were standardized to densities per 200 cm$^2$ and fourth root transformed before performing a multi-dimensional scaling analysis of Bray-Curtis dissimilarities. In addition, permutational multivariate analysis of variance (PERMANOVA) and analysis of similarities (ANOSIM) were used to test for the differences in community composition across substrate types; and a similarity percentage (SIMPER) test was used to examine which taxa contributed to those dissimilarities using Primer v6 software (*Clarke & Gorley, 2015*).

# RESULTS[1]

## Ecology of the SCB hardground macrofauna community

In this study, a total of 3,555 macrofauna individuals were counted and identified from 82 rocks collected from 231 to 2,688 m deep. Average temperatures across dives ranged from 1.80 to 7.83 °C, and oxygen concentrations ranged from 2.55 to 108.14 µmol L$^{-1}$ (Table 1). Average densities (presented as mean ± standard error) were 11.08 ± 0.87 ind. 200 cm$^{-2}$ and ranged from 0.50 to 39 ind. 200 cm$^{-2}$ (Table A1). Overall, the community was mainly dominated by the phyla Annelida (∼33%) and Echinodermata (∼28%). The other 40% of the individuals included Arthropoda (∼20%), Mollusca (∼9%), and Porifera (∼6%). Less abundant phyla (∼4% of the total) were Bryozoa (considering only branching colonies), Cnidaria, Hemichordata, Platyhelminthes, Brachiopoda, and Chordata (Fig. 3A).

In terms of morphospecies representation, these animals covered a total of 416 different taxa, excluding encrusting Bryozoa and Hydrozoa. The phylum Annelida had 169 morphospecies (∼41% of 262 morphospecies), dominating more so than for density. There were 88 morphospecies of Arthropoda (∼21% of morphospecies), 62 morphospecies of Mollusca (∼15% of morphospecies), 44 morphospecies of Porifera (∼11% of morphospecies), and 32 morphospecies of Echinodermata (∼8% of morphospecies). Less dominant phyla (∼4% of the total morphospecies) included Bryozoa (considering only branching colonies); Cnidaria; Hemichordata; Chordata; Brachiopoda; and Platyhelminthes (Fig. 3B). Macrofaunal diversity measured as H′$_{[log e]}$ on SCB rocks averaged 2.22 ± 0.07. H′ increased with increasing density (R = 0.46, $p$ = <2e$^{-16}$); at around 10 ind. 200 cm$^{-2}$, diversity stops increasing and remains relatively constant or shows little variation (Fig. 4). Notably, three of four rocks with the highest densities (>25 ind. 200 cm$^{-2}$) displayed a lower diversity than the mean (H′$_{[log e]}$ = 2.22 ± 0.07) (Fig. 4). Diversity metrics for each rock, including species richness, H′, evenness (J′), and rarefaction ES$_{(20)}$, are provided in Table A2. The surface area of the rocks was significantly correlated with the abundance of individuals (R = 0.56, $p$ = 0.39e$^{-7}$), species richness (R = 0.57, $p$ = 0.17e$^{-7}$) and H′ (R = 0.32, $p$ = 0.03e$^{-1}$).

Overall, the five most abundant taxa (∼21% of the total individuals) in the entire study were: Ophiuroidea sp. 5 (postlarvae) (304 individuals, ∼10%), *Ophiocten* cf. *centobi* (160 individuals, ∼5%), *Protocirrineris* nr. *socialis* (137 individuals, ∼4%), *Astrophiura marionae* (105 individuals, ∼3%), and Porifera sp. 5 (71 individuals, ∼2%). The following five taxa occurred on the most number of rocks: Ophiuroidea sp. 5 (postlarvae), Ophiuroidea sp. 7 (postlarvae), *Sphaerosyllis* nr. *ranunculus*, *Ophiocten* cf. *centobi* and *Munnopsurus* sp. 1. Twenty-five species (6% of the total number of species) accounted for
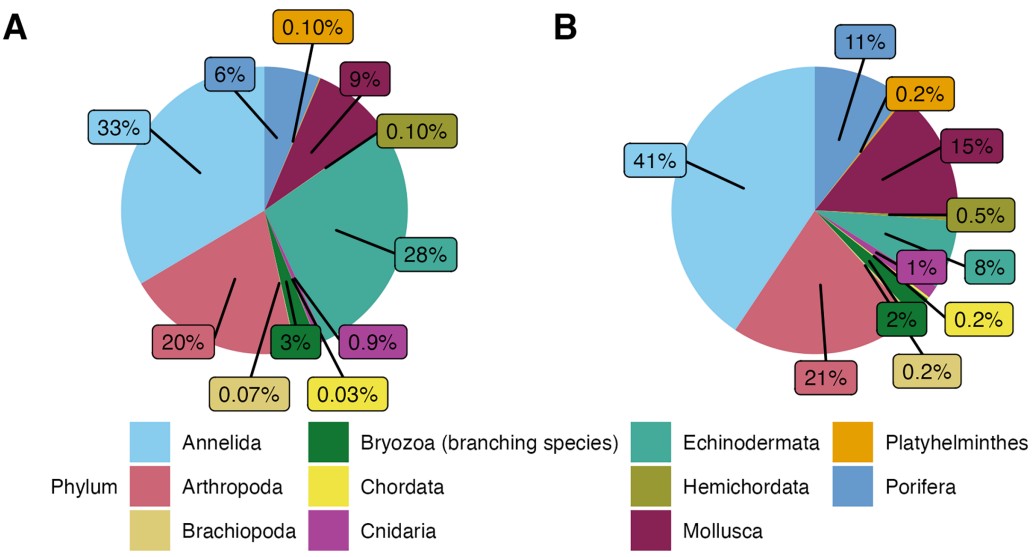

Figure 3 (A) Phyletic composition of the macrofaunal community in the Southern California Borderland (SCB) based on the number of individuals. (B) Phyletic composition of the macrofaunal community in the SCB based on the number of species. These figures do not include encrusting bryozoan colonies (455 individuals) and hydrozoans (88 individuals) since they were not identified to species level and therefore a proper comparison of individuals to species is not possible.

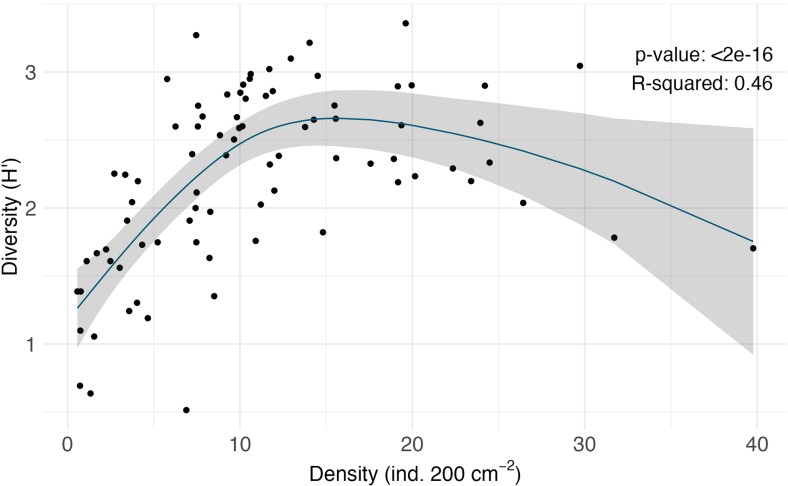

Figure 4 Non-linear regression of density (ind. 200 cm$^{-2}$) vs. diversity (H$'_{[loge]}$) of the macrofaunal communities on each rock.

over half (52.40%) of the animals collected. However, 235 of the 416 morphospecies collected (56.35%) were represented by only one or two individuals (38.60% were singletons); thus, most of the macrofaunal diversity in the hardgrounds lies with rare species. Over half of the morphospecies were found on only one rock (213 morphospecies) suggesting that much of the diversity may remain undiscovered. All 137 individuals of the third most abundant taxon, *Protocirrineris* nr. *socialis* were found on only one rock. The top ten taxa included five Ophiuroidea morphospecies, which together accounted for 21%

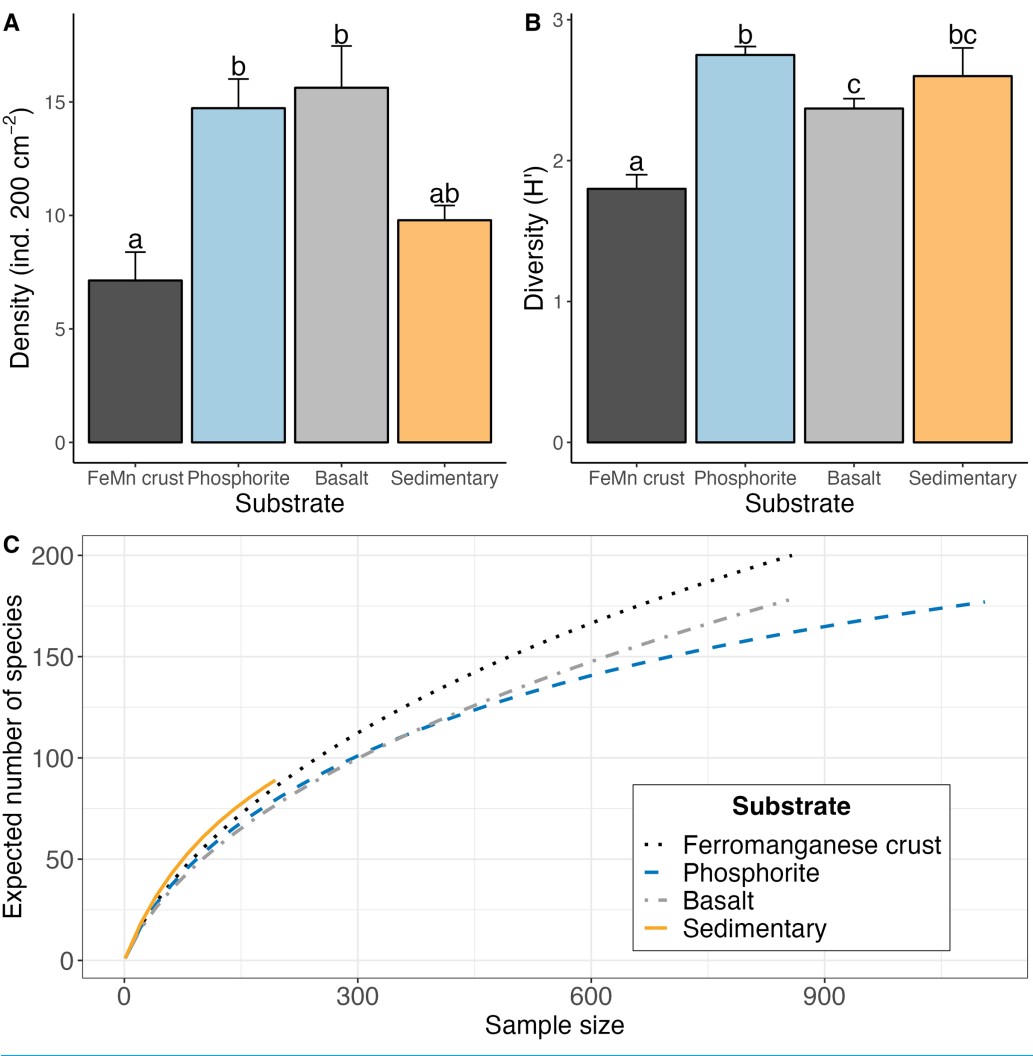

**Figure 5** (A) Average (±1 standard error) density of macrofauna per 200 cm², (B) average Shannon-Weiner diversity index (H'$_{[loge]}$) (±1 standard error), (C) rarefaction curve (ES) for macrofaunal diversity for each substrate type derived from data pooled for rocks of the same substrate type.

of all the fauna. The most frequently occurring taxa included five Ophiuroidea morphospecies, three Polychaeta morphospecies, one Isopoda, one Tanaidacea, and a branching Bryozoa. Only four morphospecies occurred on 30% of the rocks or more (>25 rocks).

## Macrofaunal relationship with substrate type

The highest average macrofaunal densities were found on basalt (15.62 ± 1.83 ind. 200 cm⁻²) and phosphorite (14.72 ± 1.28 ind. 200 cm⁻²) rocks (Chi-squared = 30.116, df = 3, $p$ = 1.30e⁻⁶). These two substrates had approximately 50% more animals than FeMn crusts (7.13 ± 1.25 ind. 200 cm⁻²) and sedimentary rocks (9.78 ± 0.64 ind. 200 cm⁻²) (FeMn crusts *vs*. basalt: z = 4.48, $p$ = <0.01; phosphorite *vs*. FeMn crusts: z = −4.49, $p$ = <0.01) (Fig. 5A). Macrofauna did not exhibit significantly different densities on

**Table 2 Average macrofaunal species richness (S), Shannon-Wiener diversity (H′), evenness (J′), and ES$_{(20)}$ for each substrate type, using rocks as replicates and substrate comparison results from the Kruskal-Wallis test.** The letters in parenthesis next to each value represent the substrates that are statistically different from one another in terms of each diversity metric.

| Substrate | Species richness (S) | Shannon index ($H'_{log_e}$) | Shannon index ($H'_{log_{10}}$) | Evenness (J′) | ES$_{(20)}$ |
|---|---|---|---|---|---|
| FeMn crust | 10.08 ± 1.48 (a) | 1.80 ± 0.10 (a) | 0.78 ± 0.04 (a) | 0.89 ± 0.02 (a) | 7.65 ± 0.62 (a) |
| Phosphorite | 23.26 ± 1.67 (b) | 2.75 ± 0.06 (b) | 1.19 ± 0.02 (b) | 0.88 ± 0.01 (a) | 12.74 ± 0.43 (b) |
| Basalt | 16.68 ± 1.24 (b) | 2.37 ± 0.07 (c) | 1.03 ± 0.03 (c) | 0.86 ± 0.02 (a) | 11.15 ± 0.53 (b) |
| Sedimentary | 17.57 ± 3.08 (b) | 2.60 ± 0.20 (bc) | 1.13 ± 0.08 (bc) | 0.94 ± 0.01 (a) | 12.78 ± 1.44 (b) |
| Chi-squared | 34.91 | 32.43 | 32.43 | 8.95 | 28.49 |
| df | 3 | 3 | 3 | 3 | 3 |
| p-value | 1.27e−07 | 4.23e−07 | 4.23e−07 | 0.029 | 2.86e−06 |

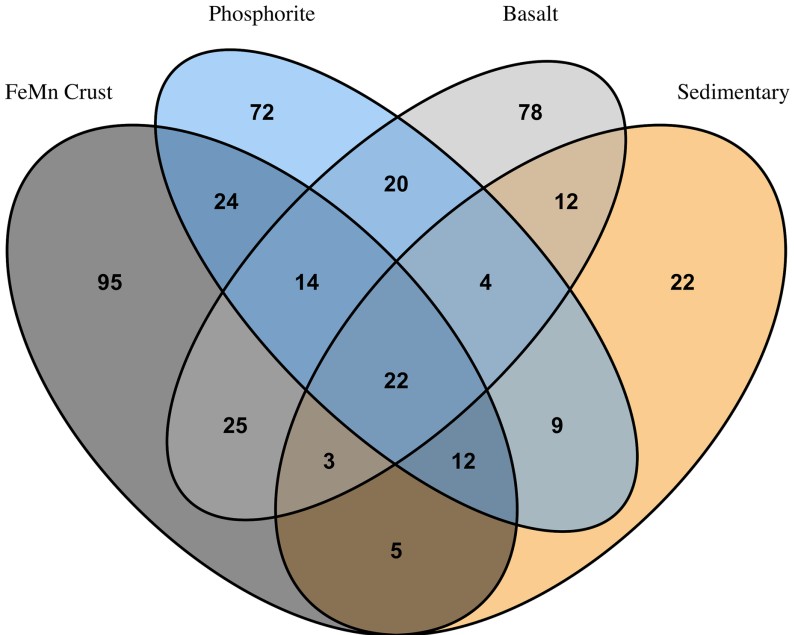

**Figure 6 Venn diagram showing numbers of overlapping invertebrate morphospecies among macrofaunal communities on different substrate types.**

phosphorite compared to basalt and sedimentary rocks, and on FeMn crusts compared to sedimentary rocks (Table A3).

Macrofaunal diversity was highest on phosphorite (avg H′$_{[loge]}$ = 2.75 ± 0.06; avg ES$_{(20)}$ = 12.74 ± 0.43) and lowest on FeMn crust (avg H′$_{[loge]}$ = 1.80 ± 0.10; avg ES$_{(20)}$ = 7.65 ± 0.62) as calculated per rock (Table 2 and Fig. 5B). Macrofaunal diversity on FeMn crust was significantly lower than on phosphorite, basalt and sedimentary rocks; and significantly higher on phosphorite compared to basalt rocks (H′: FeMn crusts *vs.* basalt: z = 2.82, *p* = 0.00; FeMn crusts *vs.* phosphorite: z = −5.31, *p* = <0.01; FeMn crusts *vs.* sedimentary rocks: z = −3.06, *p* = <0.01; phosphorite *vs.* basalt: z = −2.16, *p* = 0.02) (Table A4). All substrates exhibited similar, relatively high evenness (0.88–0.94) (Table 2).

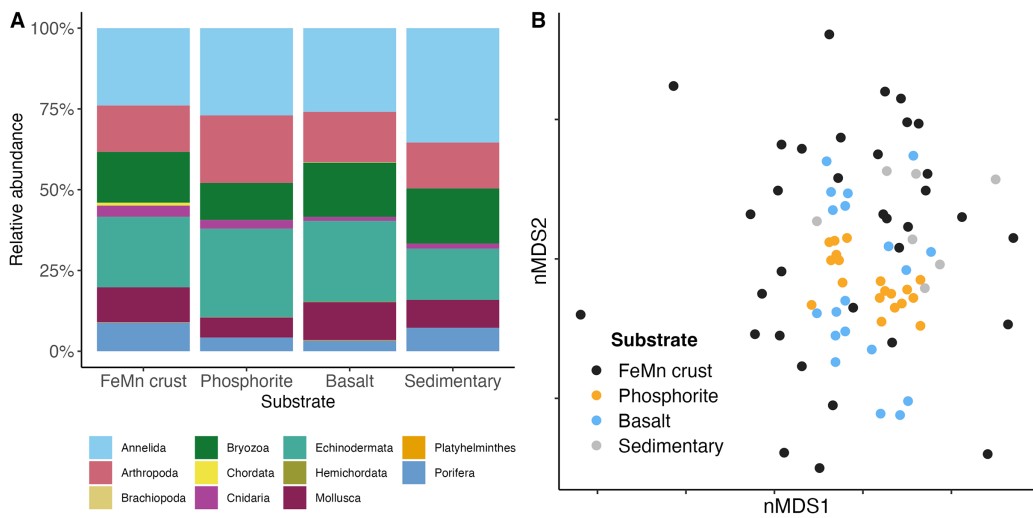

**Figure 7** (A) Community composition of macrofauna by phylum, and (B) multi-dimensional scaling analysis of macrofauna community composition across different substrate types. Each point represents the assemblage of morphospecies on a single rock.

FeMn crust macrofaunal assemblages were most similar to basalt rocks, followed by phosphorite (25 and 24 taxa in common, respectively) and least similar to sedimentary rocks (five taxa in common). When rock assemblages were pooled by substrate type, FeMn crusts and sedimentary rocks exhibit the highest rarefaction diversity (Fig. 5C). FeMn crusts had the greatest number of unique taxa (95 out of 200), followed by basalt (78 out of 178), phosphorite (72 out of 177) and sedimentary rocks (22 out of 89) (Fig. 6).

In terms of phyletic composition, the macrofaunal communities on FeMn crusts, phosphorite and basalt rocks were similar to one another. The dominant phyla across all substrates were Annelida and Echinodermata with the highest percentage of Annelida (~35%) on sedimentary rocks, and an equal proportion of Annelida (~27%) and Echinodermata (~27%) on phosphorite rocks (Fig. 7A). FeMn crusts and basalts had a similar percentage of Annelida present (~23% and ~25%, respectively). However, FeMn crusts are the only substrate with Porifera in their top ten taxa, along with three Ophiuroidea, two Amphipoda and four Polychaeta. Phosphorite rocks exhibited five Ophiuroidea, two Amphipoda, one Tanaidacea, one Isopoda, and one Polychaeta in their top ten taxa, whereas basalt rocks were the only substrate with a Bivalvia and a Holothuroidea within the top ten taxa (Table 3).

Benthic macrofaunal community composition differed across substrate type (PERMANOVA: F = 1.65, df = 3, $p$ = <0.01) between FeMn crust and sedimentary rocks (t: 1.19, $p$: 0.04), phosphorite and basalt rocks (t: 1.61, $p$: <0.01), and phosphorite and sedimentary rocks (t: 1.53, $p$: <0.01) (Table A5). Community composition also varied within a substrate, particularly among FeMn crusts (SIMPER, average similarity = 5.39%) when compared to phosphorite rocks (SIMPER, average similarity = 20.98%) (Fig. 7B). The taxa that occurred on the most number of FeMn crusts were: *Ophiocten* cf. *centobi*, *Pseudotanais* sp. 1, *Ophioleuce* cf. *gracilis*, and Ophiuroidea sp. 5 (postlarvae) (present on

**Table 3 Top 10 taxa by substrate type with corresponding percentages of total macrofaunal individuals per substrate.**

| FeMn crust | % | Phosphorite rock | % | Basalt rock | % | Sedimentary rock | % |
|---|---|---|---|---|---|---|---|
| *Ophiocten* cf. *centobi* | 6.6 | Ophiuroidea sp. 5 (postlarvae) | 12.6 | *Protocirrineris* nr. *socialis* | 14.2 | Spirorbinae sp. 1 | 5.2 |
| Porifera sp. 5 | 5.2 | *Astrophiura marionae* | 5.2 | Ophiuroidea sp. 5 (postlarvae) | 9.6 | *Ophiocten* cf. *centobi* | 5.2 |
| Ophiuroidea sp. 5 (postlarvae) | 4.4 | *Sphaerosyllis* nr. *ranunculus* | 3.3 | *Ophiocten* cf. *centobi* | 3.7 | *Sphaerosyllis* nr. *ranunculus* | 3 |
| Stenothoidae sp. 3 | 4 | *Ophiocten* cf. *centobi* | 3.2 | *Placopecten* sp. 1 | 3 | Sabellidae sp. 1 | 2.6 |
| Spirorbinae sp. 1 | 3.6 | *Amphipholis pugetana*? (juvenile) | 2.8 | *Ophryotrocha* spp. | 2.7 | *Rhachotropis inflata* | 2.6 |
| Photidae sp. 1 | 2.6 | Ophiuroidea sp. 7 (postlarvae) | 2.6 | *Astrophiura marionae* | 2.6 | Stegocephalidae sp. 3 | 2.2 |
| Serpulidae spp. (juvenile) | 2.6 | *Munnopsurus* sp. 1 | 2.5 | Ophiuroidea sp. 7 (postlarvae) | 2.2 | *Pseudotanais* sp. 1 | 2.2 |
| *Ophioleuce* cf. *gracilis* | 2.4 | *Metopa* nr. *dawsoni* | 2.2 | *Amphipholis pugetana*? (juvenile) | 1.9 | Ampharetidae sp. 2 | 1.7 |
| Sabellidae sp. 1 | 1.5 | *Stenothoe* sp. 1 | 2 | *Psolus* sp. | 1.8 | Cirratulidae sp. 1 | 1.7 |
| *Gyptis* sp. 1 | 1.4 | *Pseudotanais* sp. 1 | 2 | Bivalvia sp. 3 (juvenile) | 1.8 | Ophiuroidea sp. 11 | 1.7 |

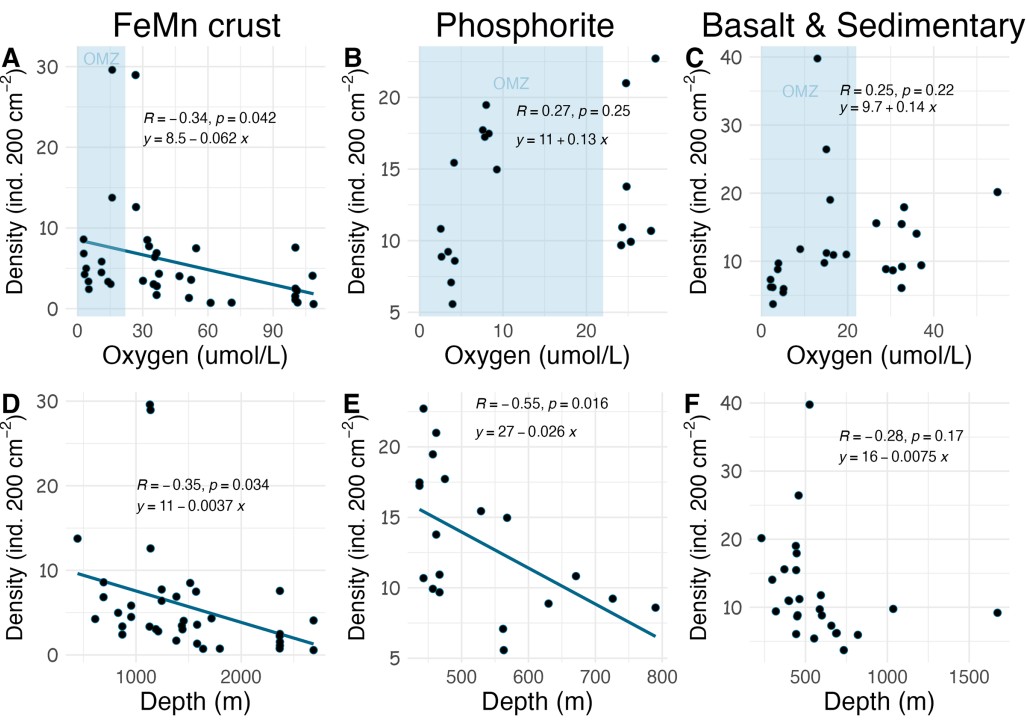

**Figure 8 Relationship between macrofaunal density and environmental variables (oxygen and depth) by substrate type (FeMn crust (A, D); phosphorite (B, E) basalt and sedimentary (C, F)).** The blue shaded areas in the top row represent the oxygen minimum zone.

12, 9, 9, and 8 out of 37 rocks, respectively). For a list of the 135 taxa that occurred only once on any FeMn crust, see Table A6. Of the 95 unique taxa on FeMn crusts, 31 taxa were Annelida, 17 taxa were Arthropoda, one taxon was branching Bryozoa, one taxon was

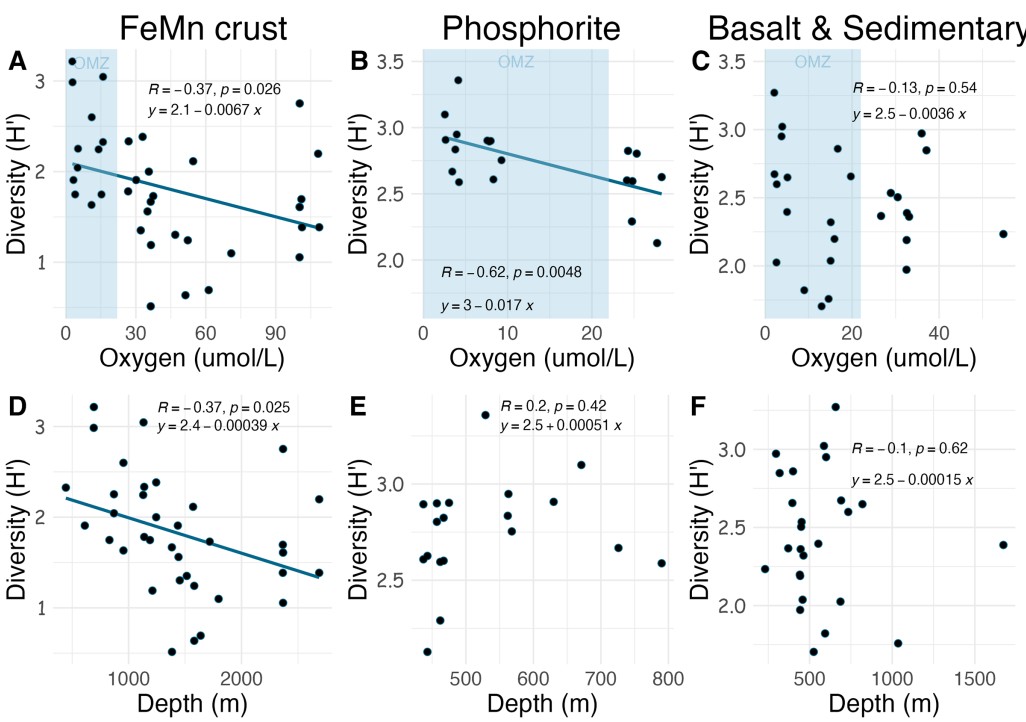

**Figure 9 Relationship between macrofaunal diversity and environmental variables (oxygen and depth) by substrate type (FeMn crust (A, D); phosphorite (B, E); basalt and sedimentary (C, F).** The blue shaded areas in the top row represent the oxygen minimum zone.

Chordata, two taxa were Cnidaria, 9 taxa were Echinodermata, 22 taxa were Mollusca, and 12 taxa were Porifera (Table A7). Of the 72 unique taxa on phosphorite rocks, 39 taxa were Annelida, 13 taxa were Arthropoda, three taxa were branching Bryozoa, one taxon was a Cnidaria, five taxa were Echinodermata, six taxa were Mollusca, and four taxa were Porifera (Table A8). The six most important taxa contributing to the dissimilarity between FeMn crust and sedimentary rock communities (SIMPER, average dissimilarity = 94.08%) were: Spirorbinae Sp. 1, *Ophiocten* cf. *centobi*, Sabellidae sp. 1, Ophiuroidea sp. 11, *Pseudotanais* sp. 1, and Ophiuroidea sp. 5 (postlarvae), which were less abundant on FeMn crusts. The six most important taxa contributing to the dissimilarity between phosphorite and basalt rocks (SIMPER, average dissimilarity = 86.47%) were: Ophiuroidea sp. 5 (postlarvae), *Sphaerosyllis* nr. *ranunculus*, *Amphipholis pugetana*? (juvenile), *Munnopsorus* sp. 1, Ophiuroidea sp. 7 (postlarvae), *Ophiocten* cf. *centobi*, which were nearly absent on phosphorite rocks. The six most important taxa contributing to the dissimilarity between phosphorite and sedimentary rocks (SIMPER, average dissimilarity = 88.14%) were: Ophiuroidea sp. 5, Spirorbinae sp. 1, *Ophiocten* cf. *centobi*, *Sphaerosyllis* nr. *ranunculus*, *Amphipholis pugetana*? (juvenile), Ophiuroidea sp. 7 (postlarvae), which were nearly absent on sedimentary rocks.

### Macrofaunal relationships with oxygen and depth as a function of substrate type

Oxygen and depth exhibited multicollinearity (r = 0.82, $p = 0.16e^{-33}$). This complicated the interpretation of relationships with the macrofauna community, as changes in one variable may be confounded by the influence of others. Therefore, the following results are presented for each substrate to test the effects of each variable among comparable samples.

Densities on FeMn crust decreased with increasing oxygen (Fig. 8A); however, on phosphorite, basalt, and sedimentary rocks (which were found at shallower depths), densities showed no relationship with oxygen (Figs. 8B, 8C). On FeMn crust and phosphorites, macrofaunal densities decreased with increasing water depth (Figs. 8D, 8E) but no trend was present for the other substrates (Fig. 8F). On FeMn crust and phosphorites, macrofaunal diversity also decreased with increasing oxygen (Figs. 9A, 9B) but no trend was observed on the other substrates (Fig. 9C). Macrofaunal diversity on FeMn crust decreased with increasing water depth (Fig. 9D); however, macrofaunal diversity exhibited no relationship on the other substrates (Figs. 9E, 9F).

## DISCUSSION[1]

### Macrofaunal relationships to substrate type

In line with previous studies on FeMn crusts on seamounts, our results from the SCB revealed lower fauna abundance on FeMn crusts compared to non-FeMn substrates (*Grigg et al., 1987*; *Schlacher et al., 2014*). In the SCB, phosphorite rocks had 50% more macrofauna than FeMn crusts on average. This trend is driven by phosphorite rocks retrieved from Coronado Escarpment, Cortes Bank and Patton Ridge South at depths <700 m. Among the surveyed sites, FeMn crust from Little Joe Seamount (~2,700 m) exhibited the lowest macrofaunal density and the highest diversity when pooled by substrate. *Vlach (2022)* reported the same pattern for megafauna, which exhibited the lowest density and highest diversity on FeMn crust at Little Joe Seamount in the SCB. On a per rock basis, FeMn crusts had lower diversity than the other substrates (Table 2). However, when data were pooled by substrate (across all rocks of the same type), FeMn crusts exhibited the highest diversity as shown by rarefaction diversity (Fig. 5C). The broader depth range from which FeMn crusts were collected likely contributed to capturing a greater diversity (Fig. 10). This mirrors the high diversity recently reported for fauna in the polymetallic nodule fields in the Clarion Clipperton Zone (*Rabone et al., 2023*).

Results presented here show that the community composition of macrofaunal assemblages on FeMn crusts in the SCB have a higher number of distinct taxa and their rarefaction diversity is higher compared to non-FeMn substrates. *Corrêa et al. (2022)* and *Schlacher et al. (2014)* also noted distinct biological communities on FeMn crusts compared to non-FeMn substrates at the Rio Grande Rise (seamount region in the Southwest Atlantic) and at a Hawaiian Seamount Chain in the Central North Pacific,

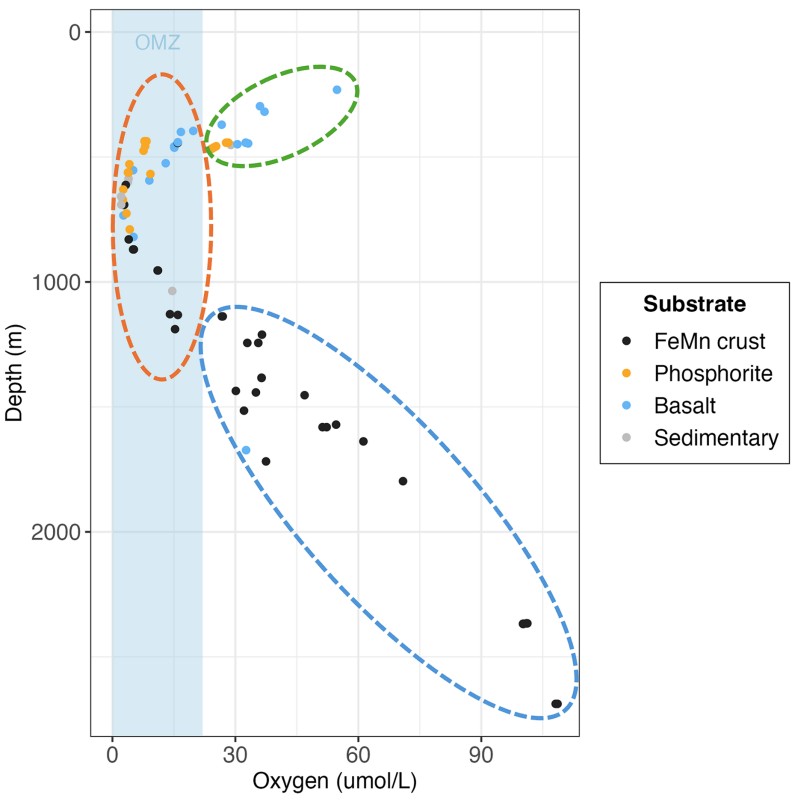

**Figure 10 Scatterplot of individual rock samples as a function of oxygen and depth at collection site, colored by substrate type.** Ellipses represent the oxygen minimum zone (OMZ) categories (green = above OMZ, orange = within OMZ and blue = below OMZ).

respectively. *Vlach (2022)* reported that megafaunal communities on FeMn crusts exhibit higher rarefaction diversity compared to phosphorite and other rock types in the SCB.

We hypothesize that microbial communities associated with FeMn crusts may vary with water depth, as shown by *Kato et al. (2018)* on a seamount in the NW Pacific (1,150–5,520 m) and *Bergo et al. (2021)* on the Rio Grande Rise (661–903 m). If this holds true for the SCB, microbial variability may drive trophic diversification among macrofaunal consumers, potentially explaining the high diversity observed in macrofaunal communities on FeMn crusts across depth ranges of 691–2,688 m. In fact, trophic diversity of macrofauna on FeMn crusts in the SCB was found to be highest when compared to those on phosphorites and other rock types (*Pereira et al., 2024*). Further research is needed to determine whether microbes are distinct across depth and substrate types in the SCB.

Variation in macrofauna community composition among FeMn crusts may also be influenced by faunal tolerance to metal concentrations, impacting their settlement (*Schlacher et al., 2014*). For instance, *Verlaan (1992)* observed higher Foraminifera densities on FeMn crusts compared to basalt rocks, although it is unclear which aspect of the rocks is supporting the macrofaunal communities. On the other hand, studies like *Veillette et al. (2007)* found no clear relationship between the geochemical composition of

FeMn crusts and associated fauna. These conflicting results suggest the need for more comprehensive research on interactions of mineral composition, microbes, and fauna (*Clark, 2011*; *Schlacher et al., 2014*). The chemical composition of FeMn crust varies with water depth, distance from shore, surface productivity, and distance from the OMZ (*Usui et al., 2017*; *Mizell et al., 2020*; *Benites et al., 2023*). For example, *Benites et al. (2023)* identified higher concentrations of certain metals, such as Mn, Co, V, As, Mo, Tl, U, Zn, and Sb in FeMn crusts collected at depths exceeding 2,000 m in the southwest Atlantic Ocean. Given the diverse depth range of FeMn crust in this study, there could be potential variations in metal concentrations among rocks. If geochemistry influences faunal distribution based on metal tolerance, it could contribute to the observed community composition differences among FeMn crusts (Fig. 7B) compared to phosphorites found at similar depths.

## Macrofaunal relationships to environmental variables

The relationship between substrate type and macrofaunal density, diversity and composition may be best explained by environmental factors occurring where each of the substrate types were collected, rather than by substrate type alone. Oxygen and depth were found to be significantly correlated in this study, and the relationship between oxygen and density and diversity of the macrofaunal assemblages is influenced by the depth categories as per a covariate statistical test. Most FeMn crusts were collected from deeper waters (>600 m) and a broad depth range. In contrast, all phosphorite rocks, and most of the basalt and sedimentary rocks were collected from shallower waters and a smaller depth range (231–800 m) (Fig. 10). FeMn crusts were the only substrate collected from the most oxygenated waters ($O_2 > 90 \ \mu mol \ L^{-1}$) at the deepest depths (>2,200 m). Most phosphorite rocks (12 out of 19) were found within the OMZ above depths of 800 meters (Fig. 10).

Across all substrates, density decreased with increasing water depth in the SCB. This trend has been observed in other deep-ocean studies from sediments (*Levin et al., 2000*; *Wei et al., 2012*; *Baldrighi et al., 2014*), presumably as a result of decreasing food availability with depth (*Ramirez-Llodra et al., 2010*). Macrofaunal density on FeMn crust was anticorrelated with oxygen and water depth; with lowest values at water depths >2,000 m and highest values within the OMZ (400–1,100 m). Decomposition of organic matter within the OMZ happens at a slower rate (*Ma et al., 2021*); thus, more food may arrive at the seafloor within the OMZ, which could explain the pattern of high density on FeMn crusts, phosphorite, basalt and sedimentary rocks in this zone. The high density of macrofauna observed in the OMZ suggests a potential resilience to low oxygen. The presence of hard substrates may offer animals increased exposure to water flow, potentially enhancing their ability to tolerate low oxygen conditions compared to their counterparts in sediments, where oxygen depletion occurs rapidly with depth into the sediment column in organic-rich sediments (*Jørgensen et al., 2022*).

*Schlacher et al. (2014)* found that faunal assemblages may vary from one site to another within a single seamount (separated by 1–2 km), between different seamounts, and due to depth variations on Pacific Ocean seamounts. The dissimilarity of macrofauna among FeMn crusts sampled in this study may thus be explained by the different sites and depth

ranges where the crusts were found (ranging from 600 to 2,700 m). To illustrate, FeMn crusts exhibited high dissimilarity in community composition at different sites on San Juan Seamount. Since all FeMn crusts were found at depths of >600 m and most of them were at >1,000 m (Fig. 10), the high diversity could be linked to low dominance due to diminishing supply of organic matter with depth (*Wei & Rowe, 2019*; *Levin et al., 2001*). *Levin et al. (2000)* also reported highest rarefaction richness for pooled samples at depths >600 m deep for deep-ocean sediments.

Unexpectedly, there was a significant negative correlation of diversity with oxygen concentration on phosphorite rocks (Fig. 9B). Instead of the commonly reported lowest species richness within OMZs (*Levin et al., 2001*), diversity of phosphorite rocks was highest within the OMZ, and as oxygen increased, diversity decreased. When considering individual rocks, average diversity metrics were highest for phosphorites compared to all other substrates analyzed in this study (Table 2). *Leduc et al. (2015)* found that macro-infaunal diversity in a phosphorite nodule ecosystem in the regions where the nodules were sitting is correlated with topographic heterogeneity and variability. Similarly, *Veillette et al. (2007)* found higher species richness on hard substrates with more complex surfaces. All the phosphorite rocks from this study were characterized by uneven surfaces, including depressions, crevices, and holes, which could explain the high macrofaunal diversity on these rocks. The other substrates studied, particularly FeMn crusts, tended to be smoother and flatter without as many depressions and crevices (Fig. 2). These findings suggest that the complexity of phosphorite rocks, compared to smoother surfaces like FeMn crusts, may play a role in supporting high macrofaunal diversity on this substrate. *Pereira et al. (2024)* noted that the crevices on phosphorites may facilitate the accumulation of particulate organic matter. This could provide a food source for deposit feeders. The higher abundance of ophiuroids and peracarid crustaceans, which are often deposit feeders, on phosphorites compared to FeMn crusts supports this hypothesis.

## Density, diversity and community composition of the SCB

Only a limited number of studies provide quantitative data for macrofauna (>0.3 mm) on hard substrates in the deep ocean; the most comparable are data for carbonate rocks. The average macrofaunal densities (ind. 200 $cm^{-2}$) in the SCB are similar to those found on carbonate rocks at inactive sites near methane seeps on the Oregon margin and on the Costa Rica margin (Table 4). However, the SCB macrofaunal densities are notably lower (<25%) than those reported for inactive carbonates at ~1,000 m off Costa Rica at Mound 11 and Mound 12 (Table 4). Relative to carbonates experiencing active seepage, average densities from the SCB were about 2% those of macrofauna on seep carbonates off Costa Rica at 1,000 m and 18% of average densities on carbonates at active methane seeps on the Oregon margin. SCB densities reported here are also much lower than on organic falls; they are 5% those of macrofauna on whale skeletons from Southern California; and 40% those of macrofauna on experimental wood deployed away from seepage in Costa Rica (Table 4). Higher macrofaunal densities are expected at active methane seeps as a result of bacterial production stimulated by the presence of methane and hydrogen sulfide. These

**Table 4 Summary of total macrofaunal densities (ind. 200 cm$^{-2}$) in the SCB and chemosynthetic environments at active and inactive methane seepage areas.**

| Location | Substrate | Seepage activity | Density | Standard error | Reference |
|---|---|---|---|---|---|
| SCB | FeMn crust | NA | 7.13 | 1.25 | This study |
| SCB | Phosphorites | NA | 14.72 | 1.28 | This study |
| SCB | Basalt | NA | 15.62 | 1.83 | This study |
| SCB | Sedimentary | NA | 9.78 | 0.64 | This study |
| Oregon margin (Hidrate Ridge North) | Carbonate | Inactive | 12.70 | 7.00 | *Levin, Mendoza & Grupe (2017)* |
| Oregon margin (Hidrate Ridge North) | Carbonate | Active | 61.50 | 31.00 | *Levin, Mendoza & Grupe (2017)* |
| Costa Rica margin (Mound 11) | Carbonate | Inactive | 46.70 | 34.9 | *Levin et al. (2015)* |
| Costa Rica margin (Mound 11) | Carbonate | Active | 112.20 | 93.90 | *Levin et al. (2015)* |
| Costa Rica margin (Mound 12) | Carbonate | Inactive | 86.10 | 73.80 | *Levin et al. (2015)* |
| Costa Rica margin (Mound 12) | Carbonate | Active | 212.90 | 125.30 | *Levin et al. (2015)* |
| Costa Rica margin (Mound Quepos) | Carbonate | Inactive | 20.20 | 16.50 | *Levin et al. (2015)* |
| Costa Rica margin (Mound Quepos) | Carbonate | Active | 89.90 | NA | *Levin et al. (2015)* |
| Costa Rica margin (Jaco Wall) | Carbonate | Inactive | 11.70 | 9.60 | *Levin et al. (2015)* |
| Costa Rica margin (Jaco Wall) | Carbonate | Active | 84.20 | NA | *Levin et al. (2015)* |
| Costa Rica margin (Mound 12) | Wood | Transition (near methane seep) | 26.00 | 14.00 | *Pereira et al. (2022)* |
| Costa Rica margin (Mound 12) | Wood | Active | 100.00 | 23.00 | *Pereira et al. (2022)* |
| Costa Rica margin (Mound 12) | Bone | Transition (near methane seep) | 233.00 | 210.00 | *Pereira et al. (2022)* |
| Costa Rica margin (Mound 12) | Bone | Active | 105.00 | 3.00 | *Pereira et al. (2022)* |
| Costa Rica margin (Mound 12) | Carbonate | Transition (near methane seep) | 57.00 | 26.00 | *Pereira et al. (2022)* |
| Costa Rica margin (Mound 12) | Carbonate | Active | 610.00 | 123.00 | *Pereira et al. (2022)* |
| Southern California (San Nicolas) | Whale skeleton | NA | 123.37 | NA | *Baco & Smith (2003)* |
| Southern California (Santa Catalina Basin) | Whale Skeleton | NA | 327.49 | NA | *Baco & Smith (2003)* |
| Southern California (San Clemente Basin) | Whale skeleton | NA | 220.09 | NA | *Baco & Smith (2003)* |

bacteria provide a primary food source for heterotrophic macrofauna at seeps, supporting a greater abundance of animals in these ecosystems (*Levin et al., 2013*, *2015*). In contrast, inactive sites at methane seeps have similar densities to those in the non-chemosynthetic environments of the SCB, as the presumed lesser microbial activity reduces the availability of food for these organisms.

In terms of diversity ($H'_{[log_e]}$), the SCB macrofauna on basalt ($H'_{[log_e]} = 2.37 \pm 0.07$), sedimentary ($H'_{[log_e]} = 2.60 \pm 0.20$), and phosphorite rocks ($H'_{[log_e]} = 2.75 \pm 0.06$) from the SCB had a higher average H′ than the inactive carbonates from the Oregon ($H'_{[log_e]} = 2.05$ at 600 m and 1.76 at 800 m) and Costa Rica margins ($H'_{[log_e]} = 1.80 \pm 0.20$)

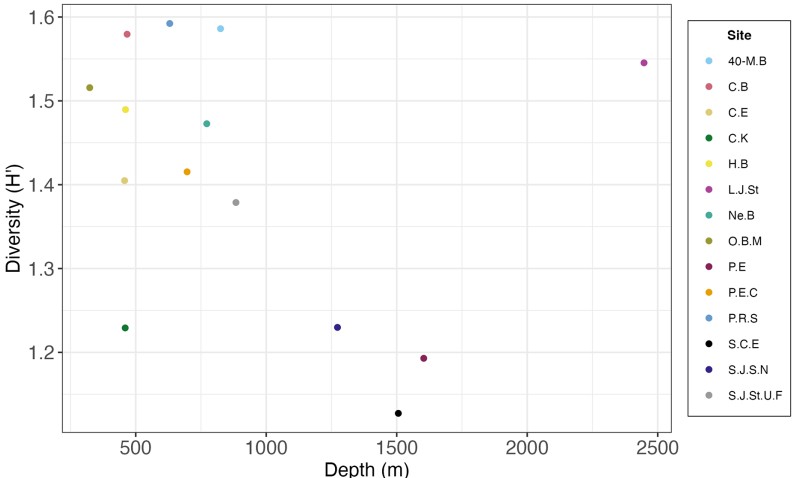

**Figure 11 Scatterplot of macrofaunal diversity (H′$_{[log_e]}$) on rocks pooled by site and average depth per site.**

and active methane seep sites off Oregon ($H'_{[log_e]} = 1.95$ at 600 m and 1.86 at 800 m) (*Levin et al., 2015*; *Levin, Mendoza & Grupe, 2017*), whereas the SCB FeMn crusts had comparable diversity ($H'_{[log_e]} = 1.80 \pm 0.10$). Diversity at active seeps across the Costa Rica margin ($H'_{[log_e]} = 2.30 \pm 0.10$) is similar to that on SCB basalt (*Levin et al., 2015*).

Non-reducing hard substrates such as those studied here and inactive carbonates in Costa Rica from 990–1,800 m deep (*Levin et al., 2015*) appear to exhibit similar proportions of annelids, molluscs (including gastropods), and arthropods. In contrast, macrofaunal communities exposed to active seepage are dominated by gastropods and annelids, which may be grazing chemoautotrophic bacteria (*Levin et al., 2015*; *Levin, Mendoza & Grupe, 2017*; *Pereira et al., 2022*). The proportion of macrofaunal species found per phylum in this study was similar to that found by *Leduc et al. (2015)* in sediments surrounding phosphorite deposits, with arthropods and annelids being the most diverse groups overall and cnidarians one of the least diverse.

## Do paradigms developed for deep-sea sediments apply to SCB hard substrates?

Most deep-sea paradigms, such as those involving depth gradients, have evolved based on the study of sediment ecosystems. This study offers an opportunity to examine these paradigms for hard substrates. Researchers initially proposed the concept of peak diversity occurring at mid to lower bathyal depths (1,500 to 2,000 m), establishing the unimodal hypothesis for deep-sea diversity (*Rex, 1981*). However, the SCB hardground macrofauna may not exhibit the typical unimodal diversity pattern observed in sediments (*Rex, 1981*). In the SCB this pattern appears to be inverted with highest diversity at shallower depths (500 to 1,000 m) and deeper depths (2,500 m), and low diversity at intermediate depths (1,250 to 1,600 m) (Fig. 11). However, this pattern should be substantiated with additional sampling below 2,000 m, as our study had only one site (Little Joe Seamount) at these deeper depths. Other studies have also countered the universal applicability of the

unimodal diversity hypothesis for deep-sea ecosystems (*Levin et al., 2001*). The SCB study also contributes to a growing understanding of the high heterogeneity and complexity of continental margins and the deep sea in general (*Levin & Sibuet, 2012*; *Danovaro, Snelgrove & Tyler, 2014*), and the California margin in particular (*Kuhnz et al., 2022*). The original paradigm of a homogeneous, desert-like sediment covered ecosystem has been replaced by one of heterogeneous substrates, topographic features, and environmental conditions supporting diverse biota. The SCB escarpments, seamounts, knolls, and ridges comprised of FeMn crusts, phosphorites, basalts, and sedimentary rocks spanning a range of depths, temperatures, and oxygen regimes reflect this heterogeneity.

One deep-sea paradigm that appears to be supported by this study involves rarity. As observed in deep-sea ecosystems (*Bax, 2011*), our SCB samples were dominated by rare species that appear as one or two individuals in the whole study. Such species (singletons or doubletons) accounted for 56.35% of all individuals sampled. This finding suggests that macrofauna in mineral-rich hard substrates may resemble macrofauna of deep-sea sediments in being comprised largely of rare species (*Carney, 1997*). This rarity trend has also been observed for manganese nodules considered for deep-sea mining in the Clarion Clipperton Zone located in the Pacific (*Christodoulou et al., 2019*; *Macheriotou et al., 2020*; *Pape et al., 2021*), and it highlights the potential loss of unique biodiversity within deep-sea ecosystems with the exploitation of associated marine minerals.

## Edge effects

According to *Levin (2003)* and *Gooday et al. (2010)*, large abundances of ophiuroids are common at OMZ edge zones due to increased food availability. Five of the 10 most abundant taxa on phosphorites, which dominated near OMZ boundaries, were Echinodermata morphospecies, including Ophiuroidea sp.5 (postlarvae) and *Astrophiura marionae*, *Ophiocten* cf. *centobi*, *Amphipholis pugetana*? (juvenile), and Ophiuroidea sp.7 (postlarvae). The most abundant Annelida species was *Sphaerosyllis* nr. *ranunculus* and the other most abundant morphospecies were members of the Isopoda, Amphipoda, and Tanaidacea orders. Ophiuroidea morphospecies reached their highest densities on phosphorite rocks at shallower depths (0–500 m), and at the lower transition zone of the OMZ where oxygen concentrations ranged from 6–22 and 22–40 $\mu$mol $L^{-1}$. Earlier studies on the margin off Central and Southern California also found high densities of ophiuroids at oxygen concentrations of 22.30 and 17 $\mu$mol $L^{-1}$ (*Smith & Hamilton, 1983*; *Thompson et al., 1985*). A similar trend was observed for the megafauna of the SCB (*Vlach, 2022*).

## Relevance for seafloor management

The seafloor of the California continental margin is vulnerable to multiple stressors, including pollution (*Schmidt et al., 2024*); warming, acidification and deoxygenation (*Evans et al., 2020*); offshore wind infrastructure; and resource extraction (bottom fishing, oil and gas extraction, and potentially seabed mining) (*Ramirez-Llodra et al., 2011*). Studies are lacking on how these different stressors interact with communities on different substrate types. The examination of all FeMn crusts revealed that in aggregate they host a high level of diversity and 47% of the species examined were exclusive to this substrate.

Similarly, phosphorite rocks exhibited notable diversity on an individual rock basis and 40% of the species found were exclusive to this substrate. Because diversity is strongly influenced by depth and substrate type, protections aimed at preserving biodiversity that cover a broad range of depths and substrates in the SCB may be more effective.

Protections against harmful commercial fishing gear have been implemented by NOAA Fisheries on San Juan Seamount, 40-Mile Bank, and Northeast Bank due to their classification as Habitat Areas of Particular Concern (HAPC) (*Pacific Fishery Management Council, 2023*; *NOAA, 2021*). HAPCs are conservation priority areas considered for their rarity, ecosystem function importance, and sensitivity to human activities (*NOAA, 2021*). Although oil and gas extraction occurs off Southern California, there is a ban on new leases in state waters, and no new federal leases have occurred off California. In addition, California state waters (0 to three nautical miles from shore) have been protected from mining under the California Seabed Mining Prevention Act since 2022 AB1832 (2022). At present, there are no mining contracts under consideration off California, and metal grades suggest it is not likely to happen soon (*Conrad et al., 2017*). However, the sites studied here are in federal waters and could be susceptible to disturbance from mining, should it ever occur. The baseline findings presented here, concerning heterogeneity, rarity, and high macrofaunal diversity of mineral-rich substrates can help us understand other regions where FeMn crust and phosphorite rocks are present and being considered for exploitation.

## CONCLUSIONS

1) This study highlights the under-researched macrofauna of Southern California Borderland (SCB) mineral-rich hardgrounds, revealing a diverse fauna with few highly dominant species and many rare taxa.

2) Macrofauna on hardgrounds in the SCB are highly heterogeneous with respect to density, diversity, and composition as a result of varied substrate type, water depth, location, and seawater oxygen concentration.

3) Ferromanganese (FeMn) crusts in the SCB exhibit high macrofaunal diversity, likely due to the heterogeneous environmental conditions at the different study sites, including varying temperatures, water depth, oxygen levels, and food supply.

4) SCB hard substrate macrofauna at bathyal depths exhibit patterns that counter several deep-sea paradigms, including the absence of a unimodal diversity-depth relationship (mid-slope maximum) and lack of depressed diversity in the oxygen minimum zone, but they resemble deep-sea sediments in having high diversity comprised of rare species.

5) Additional research is needed on (a) how and if the microbial community differs among substrate types and moderates macrofaunal community structure, through microbial settlement cues, chemical mediation, or food provision and (b) if disparity in surface texture among substrate types contributes to differences in macrofaunal density and diversity across the studied substrates.

6) Biodiversity data for the SCB reported here and observations of high heterogeneity can inform offshore resource management and conservation actions with regards to fishing, pollution, climate change, energy infrastructure and potentially seabed mining.

## ACKNOWLEDGEMENTS

We thank the captains, crews, pilots, technicians, and science parties aboard the E/V Nautilus (NA124) and R/V Falkor (FK210726) for their invaluable support at sea. Special appreciation goes to Johanna Gutleben and Paul Jensen for their significant contributions and expert advice during the expeditions. We are also thankful to Anela Choy for her insightful inputs. We acknowledge that portions of this manuscript were previously published as part of a thesis (https://escholarship.org/content/qt4tz9r4fh/qt4tz9r4fh.pdf?t=sbf1ne).

Our research benefited immensely from the expertise of taxonomic specialists Oliver Ashford, Tim O'Hara, and Charlotte Seid, who played crucial roles in identifying specimens. Additionally, we acknowledge Ailish Ullman, Ximena Flores, and Ana Patricia Galindo for their important contributions in the laboratory; and Devin Vlach for his contributions aboard R/V Falkor and E/V Nautilus. This work would not have been possible without the collective efforts and dedication of all involved. Any use of trade, firm, or product names is for descriptive purposes only and does not imply endorsement by the U.S. Government.

### Funding

This research was supported by NOAA OER awards NA19OAR110305 and NA23OAR0110520. The Ocean Exploration Trust and the Schmidt Ocean Institute funded additional ship time. There was no additional external funding received for this study. The funders had no role in study design, data collection and analysis, decision to publish, or preparation of the manuscript.

### Grant Disclosures

The following grant information was disclosed by the authors:
NOAA OER Awards: NA19OAR110305 and NA23OAR0110520.
The Ocean Exploration Trust and the Schmidt Ocean Institute Funded Additional Ship Time.

### Competing Interests

The authors declare that they have no competing interests.

### Author Contributions

- Michelle Guraieb conceived and designed the experiments, performed the experiments, analyzed the data, prepared figures and/or tables, authored or reviewed drafts of the article, and approved the final draft.

- Guillermo Mendoza analyzed the data, authored or reviewed drafts of the article, and approved the final draft.
- Kira Mizell analyzed the data, authored or reviewed drafts of the article, and approved the final draft.
- Greg Rouse analyzed the data, authored or reviewed drafts of the article, and approved the final draft.
- Ryan A. McCarthy analyzed the data, prepared figures and/or tables, and approved the final draft.
- Olívia S. Pereira conceived and designed the experiments, authored or reviewed drafts of the article, and approved the final draft.
- Lisa A. Levin conceived and designed the experiments, authored or reviewed drafts of the article, and approved the final draft.

## Data Availability

The raw data are available in the Supplemental File.

## Supplemental Information

Supplemental information for this article can be found online at http://dx.doi.org/10.7717/peerj.18290#supplemental-information.

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
