# Peer review of "Deep-ocean macrofaunal assemblages on ferromanganese and phosphorite-rich substrates in the Southern California Borderland"

_PeerJ, doi:10.7717/peerj.18290_

## Round 0.1 · original submission · Minor Revisions

Your manuscript has been carefully reviewed by two external reviewers. Both reviewers think that your work is of potential interest; however, they also raised several concerns about your study.

In particular, they have concerns about (1) the relationship between rock surface area and species diversity and abundance, (2) the uneven taxonomic resolution provided for the taxa, and (3) the lack of specific research hypotheses/questions. Please address these outstanding issues in your revised manuscript.

Reviewer 1 ·

Basic reporting

This is a welcome study to evaluate the macrofauna present on mineral-rich stones and rocks in the deep-sea environment off the California coast. The paper is generally well-written and succinct. Figures and tables overall are clear and relevant. I have no major criticisms. However, there are a few points that the authors may wish to consider:

1. Describe the rock collecting process in more detail. Given that all collections were performed by manipulators on ROVs, I would assume there are limits to the size and shape of what can be handled, and this in turn may influence the kinds of rocks that are collected. How were the rocks selected, i.e., was there a protocol developed to choose which rock to collect? Was the decision done by a particular person? Were loose ones chosen, or only those rigidly attached to other rocks, or a mixture of both? What were the size(s) of the bioboxes used?
2. Clarify the relationship between rock surface area vis-à-vis species diversity and abundance. The study suggests that substrate type is an important parameter determining macrofaunal diversity and abundance, but the size of the rock/surface area can play a part as well. I made quick scatter plots of abundance/species richness against surface area and the correlations were significant. Was rock surface area accounted for in the multivariate analyses? Does substrate type determine rock shape and/or size (this is alluded to briefly in the Discussion-line 420)? Can I also suggest in this connection that an additional column for substrate type for each rock be added to Table A1 for completeness.
3. I’m a little disappointed with the uneven taxonomic resolution provided for the taxa collected from the rocks in Tables A6–A8 (Phylum column is missing from A6). Many polychaetes and some echinoderms were identified to genus and species, but most sponges, bryozoans and molluscs were not. Does this somewhat asymmetrical approach affect the interpretation of results? For instance, identification of the gastropods even to order level (e.g., distinguishing predatory neogastropods) might help to explain differences in community structure on different rocks. Conversely, I wondered if something more could be said in terms of the feeding habits of the different groups of polychaetes found on each rock.

Other comments:
a. Reference list is incomplete (missing journal names, volume and page numbers; proper nouns not capitalised, etc.)
b. Text: Italics vs non-italics (e.g., Spirorbinae refers to a subfamily and not italicised–Table 3, rows 2 and 6; also lines 316, 323; Placopecten is a genus and should be italicised); check also Table A6–Malmgreniella, Trophoniella; please check spelling of ‘Macrarelle’
c. Figures 5, 7, 8 and 9: font size of labels A, B, C etc to be enlarged for visibility
d. Additional suggestions provided in attached pdf file

Experimental design

See Section 1 above.

Validity of the findings

See Section 1 above.

Annotated reviews are not available for download in order to protect the identity of reviewers who chose to remain anonymous.

·

Basic reporting

I provided my feedback in the uploaded PDF.

Experimental design

I provided my feedback in the uploaded PDF.

Validity of the findings

I provided my feedback in the uploaded PDF.

Additional comments

Overall, this is a very nice manuscript and I really enjoyed reading it. I only had minor comments that should be fast and easy to address.

---

## Round 0.2 · accepted · Accept

After careful assessment of the revised manuscript, I can confirm that the authors have appropriately addressed all of the reviewers' concerns. I am happy with the current version of the manuscript and believe it is now ready for publication in the journal.